# Air pollution in the places of *Betula pendula* growth and development changes the physicochemical properties and the main allergen content of its pollen

Iwona Stawoska[1], Dorota Myszkowska[2], Jakub Oliwa[1], Andrzej Skoczowski[1], Aleksandra Wesełucha-Birczyńska[3], Diana Saja-Garbarz[4], Monika Ziemianin[2]*

**1** Institute of Biology, Pedagogical University of Krakow, Kraków, Poland, **2** Department of Clinical and Environmental Allergology, Jagiellonian University Medical College, Kraków, Poland, **3** Faculty of Chemistry, Jagiellonian University, Kraków, Poland, **4** The Franciszek Górski Institute of Plant Physiology, Polish Academy of Sciences, Kraków, Poland

* monika.wandas@uj.edu.pl

## Abstract

Pollen allergy becomes an increasing problem for humans, especially in the regions, where the air pollution level increases due to the traffic and urbanization. These factors may also affect the physiological activity of plants, causing changes in pollen allergenicity. The aim of the study was to estimate the influence of air pollutants on the chemical composition of birch pollen and the secondary structures of the Bet v1 protein. The research was conducted in seven locations in Malopolska region, South of Poland of a different pollution level. We have found slight fluctuations in the values of parameters describing the photosynthetic light reactions, similar spectra of leaf reflectance and the negligible differences in the discrimination values of the $\delta^{13}C$ carbon isotope were found. The obtained results show a minor effect of a degree of pollution on the physiological condition *B. pendula* specimen. On the other hand, mean Bet v1 concentration measured in pollen samples collected in Kraków was significantly higher than in less polluted places (p = .03886), while FT-Raman spectra showed the most distinct variations in the wavenumbers characteristic of proteins. Pollen collected at sites of the increased $NO_x$ and PM concentration, show the highest percentage values of potential aggregated forms and antiparallel β-sheets in the expense of α-helix, presenting a substantial impact on chemical compounds of pollen, Bet v1 concentration and on formation of the secondary structure of proteins, what can influence their functions.

## 1. Introduction

The physiological response of plants to air pollution related to progressive urbanization may lead to an increase in pollen allergenicity [1–4]. A long-term anthropogenic pollutants exposure is the most often assessed in relation to trees growing and developing under abiotic stress. Plants growing in the natural environment are exposed to the simultaneous influence of many abiotic stress factors, which results in the development of defense mechanisms at the structural

**Data Availability Statement:** The data used in the paper, with Raman analysis results and some of the physiological parameters, were uploaded to the Open Science Framework: https://osf.io/nq86f/ (DOI number: 10.17605/OSF.IO/NQ86F).

**Funding:** The study was supported by the grant of the National Science Centre, No2016/21/N/NZ8/ 01369. Grant holder was the corresponding author: Monika Ziemianin. The FT-Raman measurements were financed by DNWZ.711.58.2022.PBU statutory research project of Pedagogical University of Krakow, Poland, led by Ph.D. Iwona Stawoska. In both cases, the funders had no role in study design, data collection and analysis, decision to publish, or preparation of the manuscript.

**Competing interests:** The authors have declared that no competing interests exist.

and functional levels. In addition to stress factors such as excess or shortage of water and/or light, mechanical factors (e.g. wind), anthropogenic factors such as air pollution by solid particles as well as sulfur and nitrogen oxides play a very important role [5–7]. Each of these stress factors mentioned directly or indirectly affect the photosynthesis process and may damage the assimilation tissues [8]. The effects of air pollution are usually quickly manifested in disturbances in the light phase reaction of photosynthesis and contribute to lowering the concentration of photosynthetic pigments in trees [9, 10]. On the other hand, one of the important indicators of long-term effects of abiotic stress factors on plants is the value of $\delta^{13}C$ discrimination in plant tissues [11, 12]. It should be remembered, however, that the ability of leaves to accumulate pollutants shows a large species variability [13].

As a result of air pollution, the overproduction of stress proteins is possible (PR-proteins, Pathogenesis Related Proteins), including the main pollen protein of silver birch [14]. Birch allergens are responsible for sensitivity in 6.4%-22.4% of the European population, provoking inhalant and cross allergy symptoms [15], and the main allergen Bet v1 recognizes specific IgE in 95% of patients [16].

One of the most important aspects related to the impact of air pollution on plants, especially in large urban areas is the increase of the plants' allergenicity, by direct and indirect influence of allergenic pollen. As the main consequence of pollution, the increase of pollen and allergens production by the plants, including stress proteins, the increase of paucimicronic particles in the atmosphere and the nitration of proteins are identified. *Ambrosia artemisiifolia*, in conditions of high concentration of $CO_2$ produces more biomass and 61–90% more pollen, facilitated by early vegetative growth in spring due to the higher temperature [17–20]. The direct effect caused by $CO_2$-induced stimulation of photosynthesis and plant growth has been also reported [21]. Moreover, plant cells in stress conditions produce defence proteins, to which many allergenic molecules belong, but this problem is more debatable. It was reported, that pollen of *Cupressus arizonica* which grows in very polluted areas expresses a higher quantity of Cup a 3 [22], while rye pollen, stressed with $O_3$ shows an increase of allergenic content and, in general, protein content [23]. On the other hand, Pasqualini et al. showed that ragweed pollen stressed by $O_3$ does not show changes in the total proteins content and in the protein profile released during the first minute of hydration nor in the expression of the allergen Amb a 1 [24]. Pollen exposure to $SO_2$ or $NO_2$ leads to higher IgE reactivity in sensitive individuals [25–27], observed by the presence of additional protein bands or the existing protein structure modifications. The pollen surface absorbs airborne particles, like metals, gases, fine PM particles, which cause changes in their shape, with an abundant release of sub-pollinic cytoplasmatic particles from starch grains, orbicules and other fragments coming from the tissues of anthers, being treated as adjuvants increasing the immune response to pollen allergens. [28–30]. Increasing ROS in airways epithelium, pollen NADPH oxidases play a fundamental role in determining an increase in inflammation caused by pollen antigens [31]. It was also indicated that allergenic pollen proteins can be nitrated by gaseous atmospheric pollutants i.a. $NO_x$ and $O_3$ which can modify an immunological response to allergenic pollen, e. g. nitration of Bet v 1.0101 induces oligomerization, which increases the allergen immunogenicity [32].

In the last decades, epidemiological studies have indicated a higher percentage of people with pollen allergy in urban areas resulted from the synergistic effect of the anthropogenic dust exposure on the natural, biological components of air [33, 34]. According to the newest report of the European Environment Agency [35] air pollution is currently the most important environmental risk to human health, after climate change. Air quality related problems, such as respiratory diseases and cardiovascular diseases are considered very serious problems by European citizens [36]. Short- and long-term exposure to air pollution can lead to reduced lung function, respiratory infections, and aggravated asthma, as well as the skin irritation,

worsening of allergic contact dermatitis symptoms and a new problem occurrence: airborne contact dermatitis. Currently, 99% of the European population is exposed to ozone, 74% to PM2,5, while 48% to PM10 above the WHO Air Quality Guidelines value [37]. Among the other pollutants, PM (particulate matter) particles are widely documented as linked with pollen allergy or asthma [38–40], as they can even cause increases in type I allergies [41] or after adhesion on pollen can formulate pollen-particle complexes and can be loaded with allergens by diffusion of proteins in the presence of water [39].

The aim of the study was to determine how the different degree of air pollution affects the physiological condition of birch (*Betula pendula* Roth) and, as a consequence, the chemical composition of pollen and secondary structure of Bet v1 protein.

## 2. Material and methods

### 2.1 Study sites

The research was conducted in seven locations in Malopolska region, South Poland (N50˚31'-N49˚09'; E19˚04'-E21˚25'), including three locations in Kraków city, **Table 1**. The tested plants grow near the air pollution measurement stations, in the places with different car traffic intensity and urbanization degrees, from large agglomeration (Kraków) to smaller cities of southern Poland (Olkusz-OL, Gorlice-GO, Trzebinia-TR). The location named as Brunary-BR, came from a forested area, was taken as the reference one due to the minimal car traffic and the lack of urban and rural buildings in the vicinity.

In order to characterize the level of pollution at the studied locations, the following pollutants data: PM10, PM2.5, and $NO_x$ were obtained from the open database of the Malopolska Inspectorate for Environmental Protection in Krakow [42].

The results obtained from the air monitoring stations, in 2017–2019 clearly indicated that in a large urban agglomeration with a heavy car traffic (Kraków—AM, RU, BA), pollution with nitrogen oxides, $NO_x$, was particularly high and remained at a similar level throughout both the winter and the early spring periods. Noteworthy is the fact that the highest $NO_x$ level was registered within one of the busiest street in Krakow, namely Aleje Mickiewicza (AM). In this location, during winter and spring periods, the level of air pollution caused by $NO_x$ was up to three times higher than the values recorded in other, also important communication routes

**Table 1. Information on the study sites, including geographical location, distance from the pollution monitoring station and short description on habitat conditions.**

| City/study site | Acronym | Co-ordinates | Distance from the pollution station | Description |
|---|---|---|---|---|
| Kraków-Aleje Mickiewicza | AM | N50˚03'; E19˚55' | 350 m | 3 specimen, close to heavy traffic road |
| Kraków-Ruczaj | RU | N50˚01'; E19˚54' | 3.000 m | More than 3 specimen, close to heavy traffic road |
| Kraków-Batowice | BA | N50˚05'; E19˚58' | 5.500 m | More than 3 specimen, at the bus terminus |
| Olkusz | OL | N50˚16'; E19˚34' | 50 m | More than 3 specimens, close to a street with little traffic |
| Gorlice | GO | N49˚39'; E21˚09' | 100 m | More than 3 specimen, the area of the estate, close to not very severe traffic road, secured with screens |
| Trzebinia | TR | N50˚09'; E19˚28' | 100 m | 3 specimen, housing estate, road with a quite heavy traffic |
| Brunary | BR | N 49˚33'; E 21˚02' | - | More than 3 specimen, forest area, 2 km away from a small village |

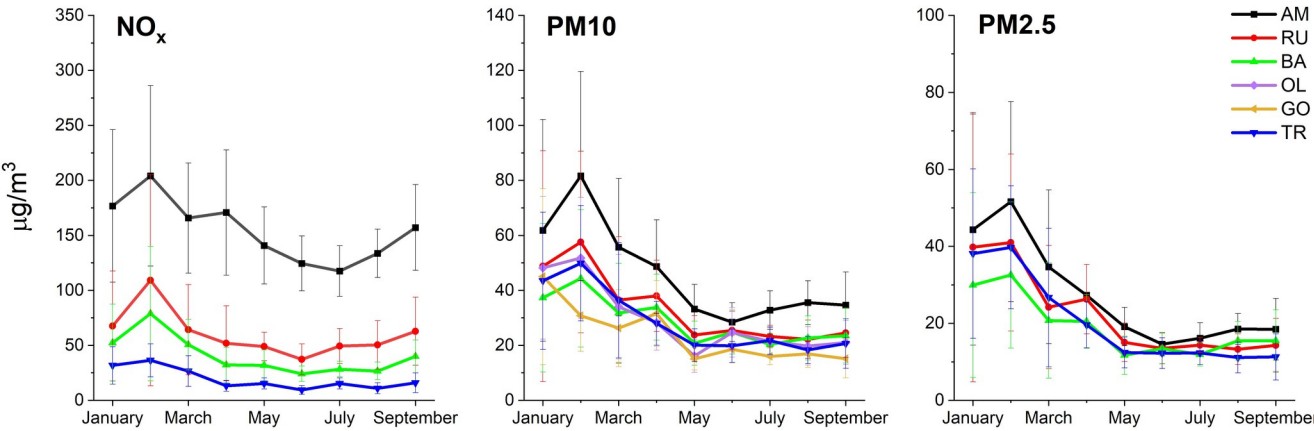

**Fig 1. Influence of the growth position in Malopolska region on the values of the parameters characterizing the physiological state of *Betula pendula* plants. A**. Changes in the maximum quantum yield of photosystem II (Fv/Fm) in leaves. **B**. Differences in the efficiency of the water splitting complex on the photosystem II donor side (Fv/F$_0$). **C**. Discrimination of $^{13}$C carbon isotope in inflorescences of *B. pendula* plants. Average values (±SD) marked with the same letters do not differ significantly according to Duncan's test, p ≤ 0,05. RU–Ruczaj, BA–Batowice, AM–al. Mickiewicza, OL–Olkusz, TR–Trzebinia, GO- Gorlice, BR–Brunary.

of the city of Krakow (BA, RU), and approximately from six to seven times higher compared to the results recorded in TR (an urban agglomeration about three times less-populated than Kraków) (**Fig 1**). Also, the concentration of suspended dust namely PM10 and PM2.5 in the AM, significantly exceeded the values recorded in others discussed locations. Based on the data collected by air monitoring stations, it can be clearly indicated that AM is the most polluted artery in Krakow.

## 2.2 Estimation of physiological condition of the trees

The physiological condition of the trees was determined by measuring the parameters of chlorophyll *a* fluorescence and parameters of light reflection of leaves. Measurements were made in mid-September in 2017–2019. It allowed to obtain a cumulative effect of environmental conditions on the physiological state of leaves. Additionally, the global impact of abiotic stress factors on trees was estimated based on discrimination value of the δ$^{13}$C carbon isotope measured in inflorescences after pollination period.

**2.2.1 Chlorophyll a fluorescence measurements.** The Chl*a* fluorescence was measured using a Hansatech Instruments (UK) Handy-PEA fluorometer according to the method of [43]. Measurements were taken in each case without damaging the leaf tissue directly on a tree, on the upper side of leaf blade. On each tree, measurements were made on 30 randomly selected leaves. The part of leaf blade on which the measurement was made was acclimated to the dark for 20 minutes using a dedicated clip. In order to excite the chlorophyll fluorescence, radiation of 3 mmol (quantum) m$^{-2}$ s$^{-1}$ was used. The measurement results were read using the PEA Plus Hansatech Instruments software (UK). Two key performance parameters of the photosynthetic apparatus were analysed: maximum photochemical efficiency of PSII (Fv/Fm) and the indicator of thylakoids structural damage (Fv/F$_0$).

**2.2.2 Leaf reflectance measurements.** The leaf reflectance spectra of *B. pendula* were measured with a CI-710 spectrometer (CID Bioscience, USA) in the range 400–1000 nm. Non-destructive measurements were made on the upper side of 30 randomly selected leaves on each tree. The signal integration time was 350 ms, the signal smoothing factor was set = 10. The results were read using SpectraSnap software (CID Bioscience, USA), then reflectance intensity curves as a function of the light wavelength were plotted.

**2.2.3 $^{13}$C discrimination.** The freeze-dried inflorescences of *B. pendula*, collected at the end of the pollen season (100 inflorescences from each site), were ground to a powder in an agate mortar. Stable carbon isotopes were analyzed on a Thermo Flash EA 1112HT elemental analyzer coupled to a Thermo Delta V Advantage mass spectrometer (Thermo Fisher Scientific, Bremen, Germany) in a continuous flow system. The samples wrapped in tin foil were burned in an oxygen atmosphere at a temperature of 1020˚C. The $CO_2$ obtained from the combustion were separated on a chromatography column and introduced directly into the spectrometer via a capillary. The measurements were calibrated according to the international standards USGS 40, USGS 41 and IAEA 600 [44]. The $^{13}$C results are presented as δ versus the VPDB standard (Vienna Pee Dee Belemnite).

## 2.3 Pollen collection and samples preparation for the further analyses

Pollen samples were collected to perform the following analyses: metabolic activity of pollen, Bet v1 concentration measurement and FT-Raman measurement. The male catkins were broken from the birch trees in all seven study sites, before the flowers were fully opened, which takes place at the turn of March and April. On average 100–150 catkins per specimen were collected. Than they were stored in a dry place for several days, to get pollen grains ready to be sieved by special sieves. Pollen was weighed, and stored in polypropylene vials at –30˚C.

## 2.4 Measurements of Bet v1 concentration in pollen

Before the immunoenzymatic measurements of Bet v1 concentration, the birch pollen collected from 60 specimen/year was extracted at 20 mg pollen/4mL in 0.1 M $NH_4$ $HCO_3$, pH 8.1, in 15 mL polypropylene tubes, after 1 min vortexing and 4 h in an endover-end rotator at 100 rpm at room temperature in the dark. The mixture was transferred to 2 mL Eppendorf vials, centrifuged for 5 min at 13,000 g, and the supernatant was collected. To 400 μl extract, BSA was added to make 0.1% w/v and the samples were lyophilized at 30˚C and stored at 4˚C until analysis. Bet v1 allergen concentration was measured using the ELISA immunoassay with the monoclonal antibodies (ELISA kit for Bet v1, Indoor Biotechnologies). Five concentrations of Bet v1 were calibrated to each plate using the Bet v1 standard reagent. As for the lyophilisate samples remaining after extraction they were dissolved in 2mL 0.1 M $NH_4$ $HCO_3$, and then analyzed in 3 dilutions (100x, 200x, 400x) in triplets. In the first stage, the plate was coated with the monoclonal antibody (4B10) and left overnight at 4˚C. Then the manufacturer instructions were followed, among others, by adding a biotinyled antibody (2B10). Bet v1 concentration was expressed as ng/10 mg of pollen.

The results were analysed according to standard statistical methods. In the first stage the normality of the distribution was tested using the Shapiro-Wilk test. The Mann-Whitney U test was applied to compare the results obtained in the samples collected in Kraków (AM, RU, BA) and outside Kraków (OL, GO, TR, BR), while the Kruskal-Wallis ANOVA was used in the case of more than two independent samples (comparison among the study sites). When the statistically significant relations were found among more than two independent samples, the appropriate post-hoc Duncan test was applied in the detailed analyses. For all these statistical tests the statistical significance was accepted at the level of $\alpha \leq 0.05$. Statistical analysis was performed using the Statistica program version 13.0 (StatSoft, Inc. 1984–2013).

## 2.5 FT-Raman spectroscopy

FT-Raman measurements were performed for (i) tablets prepared from birch pollen samples and for (ii) the liquid samples of Bet v1 protein isolated from tested pollen samples.

Prior to FT-Raman measurements, the pollen sample was defrosted, weighted (160 mg), and ground in a mortar. Next, the tablet was made using the tablet press equipment ABL& E-JASCO Polska Sp. Z o.o. (the diameter 13 mm, the pressure 200 atm). The prepared tablets were stocked in a desiccator till the time of measurement (but no longer than 4h).

The Thermo Scientific Nicolet NXR 9650 FT-Raman spectrometer equipped with a $Nd^{3+}$: YAG laser emitting at 1064 nm (the lowering of the energy of the laser radiation allows for the reduction of the fluorescence signal and as a consequence improves the quality of the spectrum) and a InGaAs detector was used to obtain the spectra. All spectra were made at an aperture of 80 and a spectral resolution of 4 $cm^{-1}$. They were collected in the range of 4000–200 $cm^{-1}$, accumulated from 128 scans (or 2000 scans for measurements performed in solution), and measured with the laser power of 0.4 W (or 0.8 W for liquid samples). Each presented result is an average of at least 6 spectra. To extract the Raman signal from the registered results (containing the fluorescence background contribution originating from the intrinsic fluorescence of plant molecules) the baseline correction was done using Thermo Scientific Omnic v.8. For further analysis of spectra, OriginPro 2020 software packages for Windows was employed.

**2.5.1 Analysis of the amide I spectral region.** To determine the secondary structural content of Bet v1, the amide I band (spectral region of 1585–1710 $cm^{-1}$) recorded for the liquid samples was decomposed using the PeakFit 4.12 (Systat Software, Inc., USA) program according to the previously presented procedure [45–47]. The analysis started with a baseline correction that used a linear function. In the next step, the second derivative of each measured spectrum was obtained, to find the number of components that build an amide I band. Finally, a mathematical algorithm, employing Gaussian and Lorentzian functions was used, to iteratively estimate parameters using the method of least squares. The areas of selected peaks correspond to their conformational contributions. The iteration procedure stopped when the best fit was achieved. For each obtained decomposition, the correlation coefficient was higher than 0.9993.

## 3. Results and discussion

### 3.1 Chlorophyll a fluorescence

The Chl*a* fluorescence parameters are considered very informative and effective in assessing the physiological condition of plants. The maximum quantum efficiency of PSII (Fv/Fm) is very often used as a reliable indicator of photosynthetic apparatus activity [48]. It has been proven that the value of $Fv/F_0$ is a more sensitive indicator of environmental stress than Fv/Fm [49].

Comparison of the maximum photochemical efficiency of PSII of *B. pendula*, Fv/Fm (**Fig 1A**) XA, shows a similar efficiency of the light-dependent processes of photosynthesis in plants growing on all sites. In all analyzed trees, the Fv/Fm parameter takes the values of 0.78–0.82, while the optimal value for plants in the vegetative state is above 0.83 [50]. Therefore, the Fv/Fm values do not indicate disturbances during light reactions in photosynthesis, caused by exposure to the negative effects of weather conditions. Greater variation between subjects was observed in the $Fv/F_0$ values (**Fig 1B**). A significant decrease of $Fv/F_0$ took place in the birch growing in Brunary (BR), which reveals some structural damage to the thylakoids, which, however, had little effect on photosynthetic electron transport within PSII [51].

### 3.2 Leaf reflectance analysis

The light reflectance measured directly on the leaf blade is strongly correlated with chemical composition, which allows an estimation of photosynthetic pigments pool in the tissue [52].

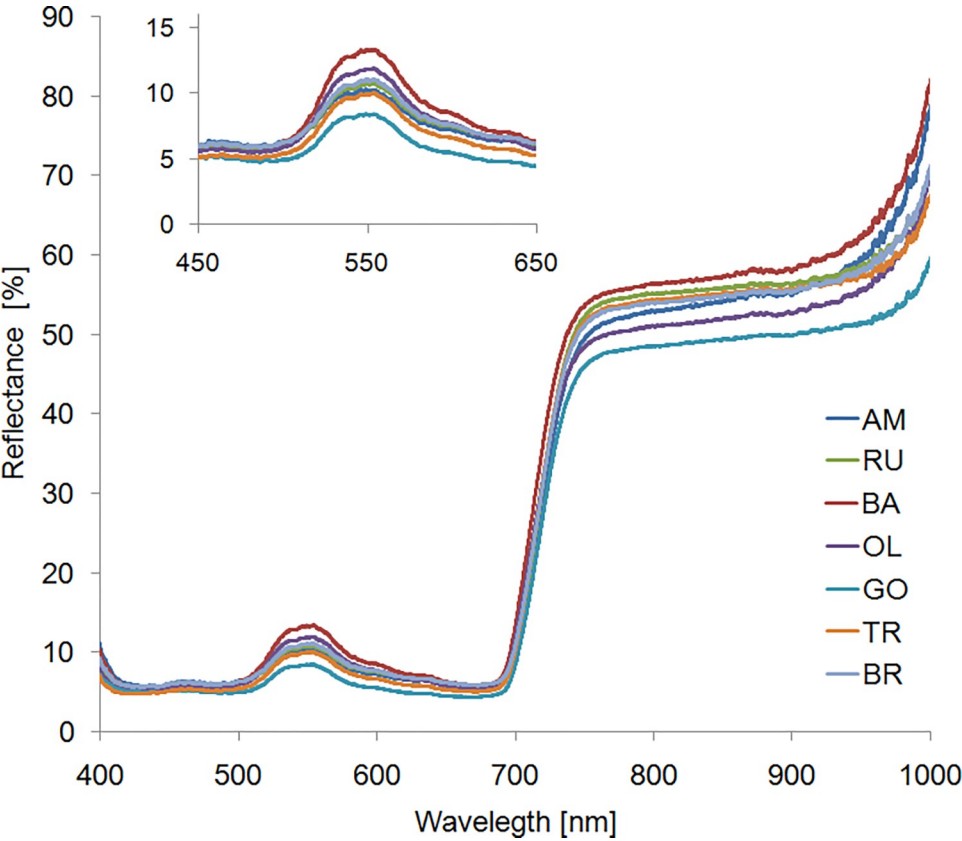

**Fig 2. Intensity of light reflectance from leaves of *Betula pendula* plants growing in positions with various pollution degrees in Malopolska.** RU–Ruczaj, BA–Batowice, AM–al. Mickiewicza, OL–Olkusz, TR–Trzebinia, GO—Gorlice, BR—Brunary.

The reflectance in the 400–700 nm range is related especially to the degree of energy absorption by chlorophylls, and to a lesser extent by carotenoids and anthocyanins [53]. In addition, this method is a good alternative to biochemical measurements, which are burdened with errors resulting from the instability of dyes [54].

The photosynthetic pigments composition of birch leaves, in particular chlorophyll content, was not significantly differentiated between the examined trees, **Fig 2**. Reflectance spectra have a similar shape in all analyzed objects. The clearly marked band in the yellow-green spectrum is the result of the chlorophyll content advantage over other pigments contained in the leaf tissue, therefore the reflectance intensity in a narrow band at 550 nm is used to indirectly assess the chlorophyll content [55].

The highest percentage values of the reflectance coefficient in the green spectrum were observed in the plant growing in BA, **Fig 2, insert**. It is associated with a reduction in the chlorophyll content, which is often observed in plants growing under abiotic stress conditions, including those exposed to air pollution [10]. On the other hand, the lowest average values of the reflectance intensity (almost half lower than in BA) were recorded in Gorlice (GO). The leaf reflectance values in PAR range in the remaining locations were intermediate and similar.

The greatest differences in the infrared part of the spectrum (700–1000 nm) were also observed between trees from BA (highest values) and GO (lowest values), **Fig 2**. The reflectance intensity in this range is mainly determined by the spatial structure of the tissues in the leaf and the ultrastructure of the parenchyma cells. The mutual position of the cell wall and

protoplasm as well as the spaces between chloroplasts and the cell wall and protoplasm play an important role [56].

## 3.3 Discrimination $^{13}$C

It is assumed that $\delta^{13}$C values for $C_3$ plants range from −20 to −35‰ (on average −27.5‰) [11, 57] In all localities, $\delta^{13}$C values oscillated near the 28.76 ± 1.24‰, which indicates that these plants were in very good physiological conditions and were able to bind $CO_2$ directly, with the predominant involvement of the ribulose-1,5-bisphosphate carboxylase oxygenase enzyme (RuBisCO), **Fig 2C**. Summarizing, the analysis of $^{13}$C discrimination showed lack of greater differences between the studied trees. Except for the site of *B. pendula* in Gorlice (GO) all differences in $\delta^{13}$C values were statistically insignificant. The largest negative $\delta^{13}$C discrimination values found for *B. pendula* growing in Gorlice indicate of optimal growth conditions in this site but it is not a magnitude indicative of any deep differences between this and the rest of the trees.

## 3.4 Bet v 1 concentration in pollen

Mean Bet v1 concentration measured in pollen samples collected within the study in Kraków (AM, RU and BA sites) was significantly higher than in less polluted places (OL, GO, TR, BR) (p = .00958). While the particular sites were considered, the significant differences were confirmed (p = .03886) (**Fig 3**). In detail, post-hoc tests indicated clear differences between the values obtained at BA (Batowice) site in Kraków and two outside sites (BR and GO) (p = .01225 and p = .03179, respectively), moreover the studied birches in Brunary (BR) produced also the significantly lower concentration of Bet v1 than at RU site in Kraków (p = .03778).

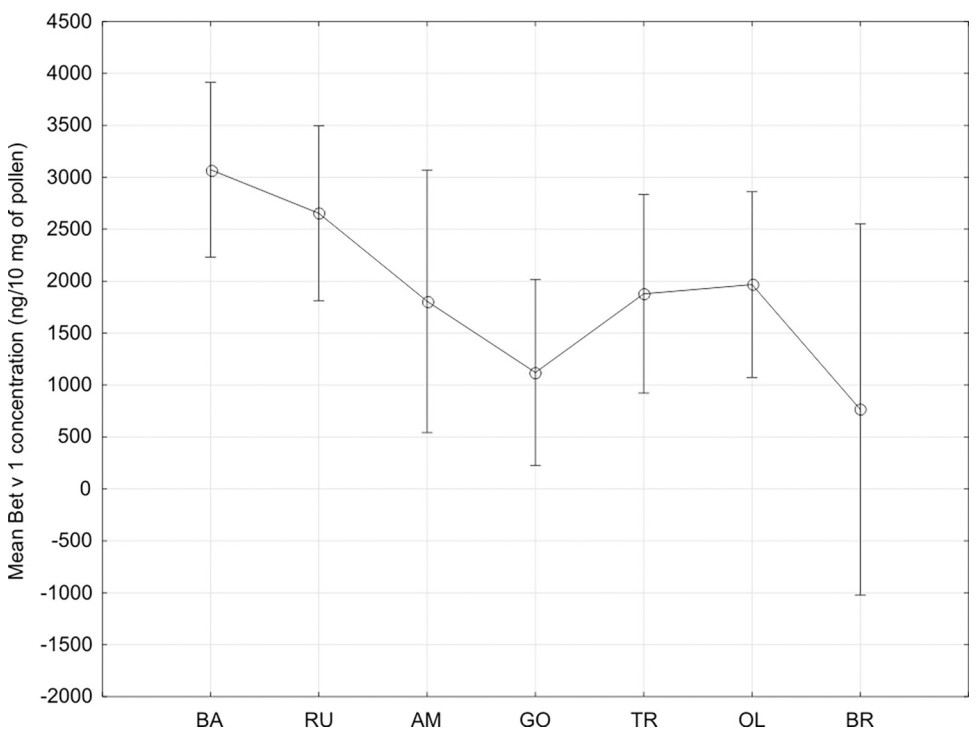

**Fig 3. Mean Bet v1 concentration in the particular study sites.** RU–Ruczaj, BA–Batowice, AM–al. Mickiewicza, OL–Olkusz, TR–Trzebinia, GO- Gorlice, BR—Brunary.

Our previous studies on the impact of PM10 on Bet v1 concentration in 20 study sites also resulted in the higher level of Bet v1 in the more polluted areas [58]. Moreover, the pollution level was the more significant factor influencing the Bet v1 concentration in comparison to the year of observations, in spite of that the birch pollen concentrations fluctuated from year to year significantly. The simultaneous exposure of pollution and *Betula* pollen is noted in April, while NOx, PM 10 and PM2.5 contamination is not as high as in winter, but still the birch pollen concentration (> 155 pollen/m$^3$) related to the risk of asthma dyspnea and PM10 concentration > 50 μg/m$^3$ is detected [59]. The similar study on Pla a1 concentration (the main allergenic protein of *Platanus hybrida* pollen), performed in two urban localities in Spain, also showed the seasonal differences, which were inversely proportional to the total pollen protein biomass [60]. [61] reported the year-to-year differences in the expression of some of *Ailanthus altissima* pollen proteins. The authors underlined that the implementation of allergomic tools for the safety assessment of newly introduced and invasive plant species would help in the comprehensive monitoring of proteomic and transcriptomic alterations involving environmental allergens.

In the case of birches, another physiological aspect should be considered, while the allergenic pollen production is discussed. It is surprising, that the lowest Bet v1 concentration was obtained in the forest site, however trees growing in this locality showed the weakest physiological conditions. Among the abiotic stress factors, those of the higher impact in the urban areas and the other ones dominated in the forest area can be distinguished. The authors suppose that in the location far away from the urban areas, the physical factors, like wind intensity, longer snow cover, lower temperature, ground water level play a greater role as the environmental factors influencing the physiological plants condition. Among others, cold stress alters the expression of putative cold responsive genes coding for an array of important proteins, like enzymes involved in respiration and the metabolism of carbohydrates, phenylpropanoids, lipids, antioxidants, what leads to subsequent production of specific proteins during cold tolerance and plays an important role in the distribution and survival of plants [62]. On the other hand, heavy metals and other pollutants contamination interfere with the growth and physiology of plants, by reducing the number of mitochondrial cristae leading to impaired oxidative phosphorylation, promoting the aggregation and condensation of chromatin as well as impaired replication and transcription, by enhancing production and accumulation of ROS, and indirectly provoking the potential changes in the pollen proteins structures [63].

## 3.5 FT-Raman spectra registered for tablets of selected pollen of Betula pendula and for Bet v1 protein solutions

In order to evaluate the influence of air pollution on the possible increase in the allergenicity of birch proteins, in particular Bet v1, which is considered to be the main birch pollen allergens [64], experiments using the FT-Raman spectroscopy technique were also carried out. Firstly, the biochemical components present in pollen collected from trees growing in selected locations were characterized, **Table 2**. The untaken measurements were performed on tablets obtained from 160 mg of each selected pollen.

FT-Raman spectra obtained for birch pollen contain information on the major chemical constituents of tested material: lipids, carbohydrates, proteins and other biopolymers like grain wall biopolymers, namely sporopollenins, and cellulose. **Fig 4** shows the specific bands registered, whereas chemical compounds assigned to the particular vibrations shown in the FT-Raman spectra are presented in **Table 2**. All obtained Raman spectra possess the fingerprint region between 800–1100 cm$^{-1}$ that is typical for C-C skeletal vibrations coming from lipids and fatty acids [65]. Moreover, for these types of components CH$_2$ wagging vibrations at

**Table 2. The characterization of chemical compounds assigned to the particular bands in the FT-Raman spectra.** The spectra were measured on the tablets of pollen of selected trees of *Betula pendula*.

| Wavenumber (cm$^{-1}$) | Components | Vibrations | References |
|---|---|---|---|
| 800–1100 | Lipids, fatty acids | $\nu$(C-C) | [65] |
| 940 | polysaccharides (amylose and amylopectin) | Skeletal modes | [65, 68, 76, 77] |
| 1124 | Polysaccharides (cellulose), disaccharides (sucrose) | $\nu_{sym}$(C-O-C) | [65, 69, 77, 78] |
| | | $\nu$(C-O)+$\nu$(C-C) | |
| 833, 852, 1171 | Sporopollenins/Tyr | Aromatic ring vibrations | [66, 67, 70–72, 77] |
| 1165 | Tetraterpenes | $\nu$(C-C) | [65, 73, 74] |
| 1305 | lipids, fatty acids; | CH$_2$ wagging vibrations | [65, 66, 77] |
| 1440 | lipids, fatty acids | CH$_2$deformation | [65–67, 77] |
| 1601 | phenolic compounds (sporopollenins) | aryl ring stretching vibrations | [65, 71, 74] |
| 1652 | proteins | C = O stretch, amide I | [45, 65, 75, 77] |

1305 cm$^{-1}$ as well as the broad band at 1440 cm$^{-1}$assigned to CH$_2$ deformation vibrations are observed [65–67]. Signals related to carbohydrates (amylose and amylopectin) are detected at 940 cm$^{-1}$, whereas the one that is linked to the presence of cellulose from the grain wall (belonging to stretching symmetrical vibration of C-O-C) is observed at 1124 cm$^{-1}$ [65, 68, 69]. Raman bands at 833, 852 and 1171, 1601 cm$^{-1}$are fitting to sporopollenins, biopolymers found in the outer wall of pollen grain [66, 67, 70, 71]. These biopolymers are based on phenylpropanoids, thus the Raman spectra possess bands associated with the vibration of aromatic rings [72]. In addition to these signals, C-C stretching vibrations at 1165 cm$^{-1}$, that belongs to carotenoid structures are observed [65, 73, 74]. Carotenoids have been reported to be commonly involved in pollen pigmentation, mainly in the exine layer [67]. Furthermore, all registered spectra possess protein band localized at around 1652 cm$^{-1}$ (amid I) [45, 65, 75].

We also pointed out the Raman bands related to the presence of selected aromatic amino acids, namely: Tyr (643, 833, 852 cm$^{-1}$), Phe (1005 cm$^{-1}$), His (1260–1265 cm$^{-1}$), Trp (1361 cm$^{-1}$), **Fig 4**. Based on the analysis of the position of the bands characteristic for Tyr molecules, one can conclude about conformational changes of these residues. For Tyr molecules, the bands' intensity ratio I (852 cm$^{-1}$)/I (833 cm$^{-1}$), of the tyrosine doublet, depends on the interactions between the -OH groups and the environment in which they are located. For tested pollen samples collected from trees growing in various urban agglomerations, the intensity of the band at 852 cm$^{-1}$ dominates over the band at a lower frequency. A slight increase in the intensity ratio I(852 cm$^{-1}$)/I(833 cm$^{-1}$) for the samples was observed, starting from the lowest to the highest value, in the order from BR(0.9894), AM(1.0036), RU(1.0061), OL(1.0112), BA (1.0304), TR(1.0319) to GO(1.0546), respectively. The increase in the intensity ratio of the tyrosine doublet, in general, indicates a gradual exposure of Tyr residues to the surface of the protein complex and a slow change of the character of the hydrogen bonds from intramolecular to intermolecular, which in turn may result from denaturation and/or aggregation of the protein. However, the values of the intensity ratio I(852 cm$^{-1}$)/I (833 cm$^{-1}$) obtained for the pollen collected from urban and non-urban areas are in the range of 0.9–1.43, and thus, according to the literature data, detected tyrosine residues could act as both donors and proton acceptors [79]. Concluding, FT-Raman spectroscopy measurements did not show that air pollution affects the location of Tyr residues in the pollen's proteins.

Raman spectra show bands typical for oscillations of the His residues. These amino acids can occur in proteins in two different tautomeric forms. The presence of the band in the range of 1260–1265 cm$^{-1}$ indicates that for the tested birch pollen we deal only with a tautomer containing hydrogen at the N3 atom, or that its predominance in quantity over the tautomer containing hydrogen at N1 atom is significant. The band at *ca*. 1005 cm$^{-1}$ is associated with the

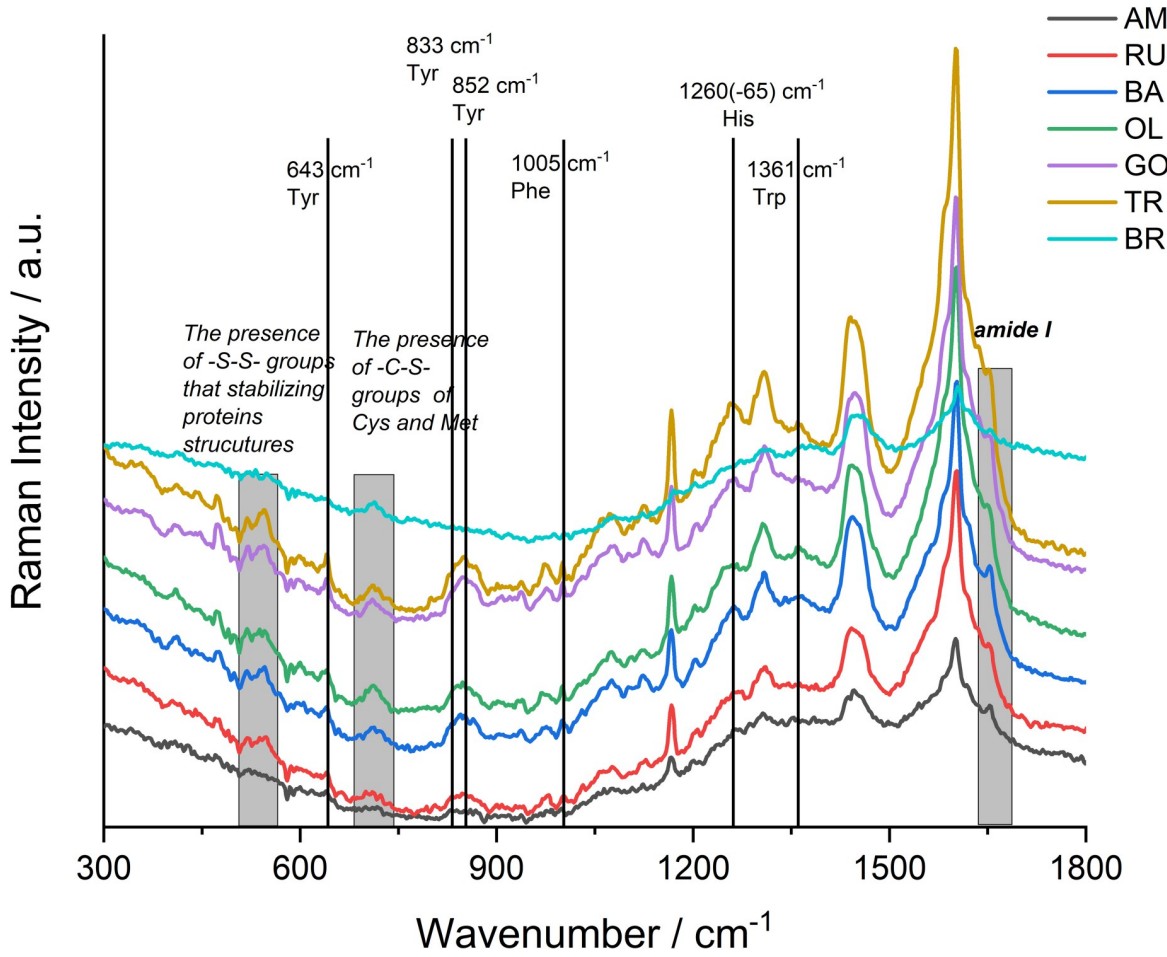

**Fig 4. FT-Raman spectra of pollen samples of selected *Betula pendula* from various localizations in Malopolska region.** RU–Ruczaj, BA–Batowice, AM–al. Mickiewicza, OL–Olkusz, TR–Trzebinia, GO- Gorlice, BR—Brunary. The wavenumbers of aromatic amino acids, as well as typical bands from protein structures, are pointed. Each spectrum is an average of at least 6 separate measurements. SD < 5%.

presence of Phe and becomes visible if Phe content in the protein structures exceeded 1%. The peak at 1361 cm$^{-1}$ is related to the presence of Trp residues in the protein structures. On the basis of the obtained spectra, it is difficult to discuss the position of Trp in the space of protein globules. It should be noted, that for samples collected at AM and RU the discussed band has the lowest intensities, which indicates Trp exposure to the hydrophilic environment and possible denaturation and/or aggregation of proteins of birch pollen collected from polluted regions. Moreover, in **Fig 4**, the bands originating from disulfide bridges, at 490–560 cm$^{-1}$, that stabilise the protein structures and the others, at 630–730 cm$^{-1}$, associated with the presence of -C-S- groups derived from Met and Cys residues are also visible.

The changes in the protein components, mentioned above can result in the protein profile, not only in relation to the main allergen Bet v1, but also in the case of other particles, like Bet v2 (profilin, actin-binding protein), Bet v3 and Bet v4 (polcalcin-like proteins), Bet v 7 (cyclophilin) and Bet v8 (glutathione-S-transferase). Special attention should be paid to the isoflavone reductase-related protein (Bet v6), being plant defence proteins and belonging to the IFRs family, which can be induced by plant stress.

In the next stage of the research, we isolated the Bet v1 protein, which is suspected of causing the strongest allergenic effect, from pollen samples. The obtained protein was transferred

into an aqueous solution and subjected to FT-Raman analysis. It is worth adding that the FT-Raman spectroscopy method, using the Raman effect, successfully competes with FTIR spectroscopy, as it has a wider application and does not require drying the samples. Therefore, it would be impossible to perform FTIR measurements on liquid (aqueous) samples of Bet v1 protein due to the limitation of the IR method.

The aim of the experiment was to study the influence of air pollution on changes in the secondary structure of Bet v1, which certainly has an impact on the protein functions and properties (including allergenic properties). From the registered FT-Raman spectra of Bet v1, the amide I band region was selected (1585–1710 cm$^{-1}$), and the decomposition was performed in order to evaluate the content of the individual secondary structures of the protein. The obtained results are shown in **Fig 5** and in **Table 3.**

The experimental profiles are represented by dashed black lines and the calculated ones by solid red lines. The calculated profiles in each panel were determined as the sums of the curve-fitted components, namely β-sheet (1636–40 cm$^{-1}$—yellow), α-helical (1650–53 cm$^{-1}$—magenta), RC (1662–66 cm$^{-1}$—brown), β-turn (1675–81 cm$^{-1}$—green) and antiparallel β-sheet (1684–96 cm$^{-1}$—blue) structures. Each curve-fitting analysis was done on the average spectrum obtained from 3 separate measurements, SD < 5%.

As a result of the performed decomposition of the amide I band, five components were obtained, which allow for the identification of various forms of secondary structures of the Bet v1 protein [80–83], **Fig 5**. In all spectra, additional bands were distinguished, with a maximum at approximately 1595 cm$^{-1}$ and in the ranges 1607–1610 cm$^{-1}$ and 1621–1627 cm$^{-1}$. They can

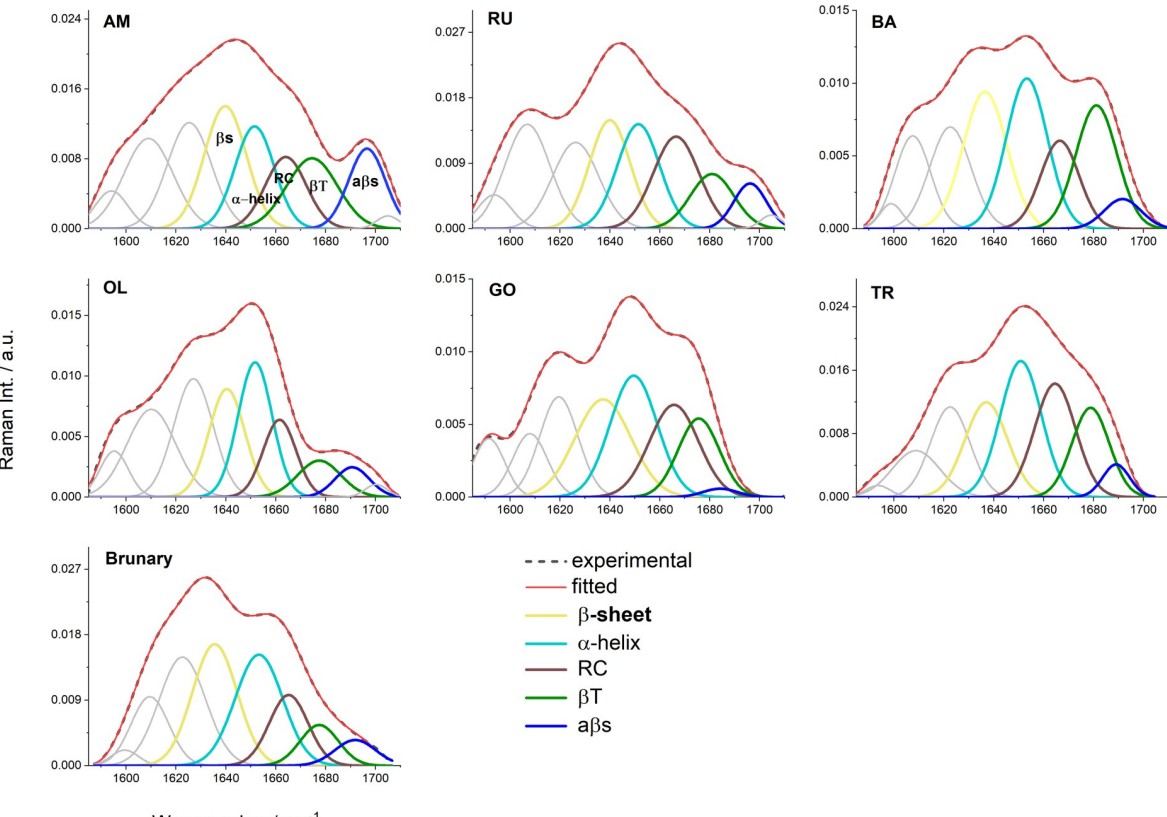

**Fig 5. Curve fitting of the FT-Raman spectra in the range of amide I obtained for Bet v1 protein, deriving from the studied areas.**
RU–Ruczaj, BA–Batowice, AM–al. Mickiewicza, OL–Olkusz, TR–Trzebinia, GO- Gorlice, BR–Brunary.

**Table 3. The quantitative estimation [%] of secondary structures content of Bet v1 protein isolated from pollen of *Betula pendula*, collected from trees growing in particular study site in Malopolska region.**

| Secondary structures of protein | Study sites/ Values [%] of secondary structures content of Bet v1 | | | | | | |
|---|---|---|---|---|---|---|---|
| | AM | RU | BA | OL | GO | TR | BR |
| β-sheet (1636–40 cm$^{-1}$) | 28 | 26 | 28 | 28 | 28 | 22 | 34 |
| α-helix (1650–53 cm$^{-1}$) | 21 | 26 | 29 | 34 | 30 | 30 | 33 |
| RC (1662–66 cm$^{-1}$) | 16 | 25 | 15 | 19 | 24 | 25 | 17 |
| β-turn (1675–81 cm$^{-1}$) | 19 | 14 | 23 | 11 | 17 | 18 | 10 |
| aβ-sheet (1684–96 cm$^{-1}$) | 16 | 9 | 5 | 8 | 1 | 5 | 6 |

be identified as the aromatic amino acid side-chain vibrations characteristic of Tyr, Phe and Trp molecules [77, 83–88]. The β-sheet structures are found at frequencies 1636–1640 and 1684–1696 cm$^{-1}$, with the latter being typical for structures rich in intermolecular hydrogen bonds (antiparallel β-sheets, aβ-sheets). β-turns can be detected at 1675–1681 cm$^{-1}$, whereas α-helical forms are found in the range of 1650–1653 cm$^{-1}$. One can also identify the band at *ca*.1662-1666 cm$^{-1}$ which is connected with random coil and/or undefined structures. Decomposition of the amide I band of the Bet v1 protein isolated from birch pollen samples showed a significant increase of β-turns (1675–81 cm-1) for all samples deriving from urbanized localization (except for OL) and an increase of antiparallel β-sheets (1684–96 cm$^{-1}$) contents for AM, RU and OL in comparison with BR–the control sample from the non-urbanized area. An increase in the random coil content for RU, OL, GO and TR compared to BR was also found. Conversely, we detected a decrease of β-sheets (1636–40 cm$^{-1}$) for all tested samples and a decrease of α-helix contents (~1650 cm$^{-1}$), especially for AM and RU (samples deriving from the most polluted regions).

In FT-Raman spectroscopy, many forms of β-structures give separate bands and their position depends on the strength of hydrogen bonds between strands [89]. Aggregated forms of β-sheets appearance are usually connected with a decrease of β-sheet content and the existence of β-structures with strong intermolecular hydrogen bonds in which antiparallel β-sheets are formed [90]. At the same time one can remember that applying stress factors, like air pollutants, may lead to the loss of an internal hydrogen bonding between amino acid residues, and in consequence, it can cause destabilization of secondary structures and protein unfolding [91]. The band observed at 1607–1610 cm$^{-1}$, besides its evidence typical for aromatic amino acids, is also categorized in the literature as the one that can be connected with aggregated structures formation [92, 93]. Thus, their increasing contents observed for AM, RU and OL may relate to aggregated structures formation, which is also online with the increasing content of aβ-sheets found for the same samples.

Our experiment has shown that air pollution resulted in the decrease of regular α-helix and β-structures and a pronounced increase of the bands characteristic for β-turns (1675–81 cm$^{-1}$) together with those at the highest frequency (1684–94 cm$^{-1}$), manifesting a formation of antiparallel β-sheet structures. The appearance of random coil structures may be a result of intermediates in the way of aggregated structures formation. The substantial increase of aggregated-like forms, connected with aβ-sheets formation, was observed for the AM and RU and for OL. Two of them (AM and RU) came from a large urban agglomeration, Kraków city, and, as we pointed out above, AM is the most polluted communication artery in Kraków.

## 4. Conclusions

Diverse growth conditions, including the degree of environmental pollution, have a minor effect on the physiological condition of individual specimens of *B. pendula*, as evidenced by

slight fluctuations in the values of parameters describing the photosynthetic light reactions, similar spectra of leaf reflectance as well as negligible differences in the discrimination values of the $\delta^{13}$C carbon isotope in plants from all sites. Therefore, changes in pollen proteins are not directly related to the physiological condition of the plant assessed by the efficiency of photosynthesis and the pigment composition of leaves.

Detailed analysis of the obtained FT-Raman spectra allows us to conclude that the most distinct variations, which can be a result of air pollution and urbanization, are observed in the wavenumbers characteristic of proteins, indicating their involvement in these processes. Pollen collected at AM and RU sites, of the increased $NO_x$ and suspended dust concentration, show the highest percentage values of potential aggregated forms and antiparallel β-sheets at the expense of α-helix, compared to BR—samples from not-urbanized areas. We found that urbanization and air pollution affect the formation of particular types of secondary structures, namely β-sheets and α-helical structures, and consequently, can influence the functions of proteins. Finally, this can lead to an increase in the allergenicity of proteins and to a more frequent incidence of allergies in sensitized individuals. Moreover, the higher Bet v1 concentration in the samples from the more polluted places was obtained.

The results may explain the failures in the treatment of people with pollen allergy, living in polluted areas, which should be considered by physicians, especially during the birch pollen seasons. Moreover, when greenery in the cities is planned, allergenic trees should not be planted, because, despite the fact that they can physiologically adapt to the local environment, they produce more stress proteins, of a higher potential allergenicity.

## Supporting information

**S1 Graphical abstract.**
(TIF)

## Acknowledgments

Authors thank also to M.Sc. Martyna Kraińska from Jagiellonian University in Kraków, Poland, Faculty of Chemistry, for performing Raman spectroscopy measurements for liquid samples.

## Author Contributions

**Conceptualization:** Dorota Myszkowska, Andrzej Skoczowski.

**Formal analysis:** Dorota Myszkowska, Jakub Oliwa, Aleksandra Wesełucha-Birczyńska, Monika Ziemianin.

**Funding acquisition:** Dorota Myszkowska, Monika Ziemianin.

**Investigation:** Iwona Stawoska, Dorota Myszkowska, Jakub Oliwa, Andrzej Skoczowski, Aleksandra Wesełucha-Birczyńska, Diana Saja-Garbarz, Monika Ziemianin.

**Methodology:** Dorota Myszkowska, Andrzej Skoczowski.

**Project administration:** Dorota Myszkowska, Monika Ziemianin.

**Supervision:** Dorota Myszkowska, Andrzej Skoczowski.

**Visualization:** Iwona Stawoska, Dorota Myszkowska, Jakub Oliwa.

**Writing – original draft:** Iwona Stawoska.

**Writing – review & editing:** Iwona Stawoska, Dorota Myszkowska, Jakub Oliwa, Andrzej Skoczowski, Aleksandra Wesełucha-Birczyńska.

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
