## [Decision Letter · Decision Letter 0]

1 Sep 2022

PONE-D-22-14561

Possible relationship between the physiological conditions of Betula pendula and physicochemical properties of its pollen at selected sites in Małopolska region

PLOS ONE

Dear Dr. Ziemianin,

Thank you for submitting your manuscript to PLOS ONE. After careful consideration, we feel that it has merit but does not fully meet PLOS ONE’s publication criteria as it currently stands. Therefore, we invite you to submit a revised version of the manuscript that addresses the points raised during the review process.

We look forward to receiving your revised manuscript.

Kind regards,

Umakanta Sarker

Academic Editor

PLOS ONE

https://journals.plos.org/plosone/s/file?id=ba62/PLOSOne_formatting_sample_title_authors_affiliations.pdf".

“The study was supported by the grant of the National Science Centre, No2016/21/N/NZ8/01369.

Grant holder was the corresponding author: Monika Ziemianin”

“The study was supported by the grant of the National Science Centre, No2016/21/N/NZ8/01369.”

“The study was supported by the grant of the National Science Centre, No2016/21/N/NZ8/01369.

Grant holder was the corresponding author: Monika Ziemianin”

Additional Editor Comments:

There are several typographical mistake in the text and table. Address then accurately. For instance, minute symbol of table, ml, µl, space between number and degree symbol, etc.

Reviewers' comments:

Reviewer's Responses to Questions

**Comments to the Author**

1. Is the manuscript technically sound, and do the data support the conclusions?

Reviewer #1: Yes

Reviewer #2: Yes

Reviewer #3: Yes

2. Has the statistical analysis been performed appropriately and rigorously? 

Reviewer #1: No

Reviewer #2: Yes

Reviewer #3: Yes

3. Have the authors made all data underlying the findings in their manuscript fully available?

Reviewer #1: No

Reviewer #2: Yes

Reviewer #3: Yes

4. Is the manuscript presented in an intelligible fashion and written in standard English?

Reviewer #1: Yes

Reviewer #2: Yes

Reviewer #3: Yes

5. Review Comments to the Author

Reviewer #1: This paper describes a study about the possible relationship between the physiological conditions of Betula pendula and the air pollution. This is a rather relevant contribution to the better understanding of the problem of increasing allergy of pollen in polluted atmospheres. A particularly important feature of this study is the acquisition of primary experimental data, which is not common. I found only one problem with the analysis of the results. Indeed, because we have a multivariate data set, and besides the univariate analysis that has been presented in the paper, a unsupervised multivariate data analysis should be done, for example principal components analysis, non-linear maps, etc. This would improved the data analysis and interpretation of the results.

Reviewer #2: The manuscript "Possible relationship between the physiological conditions of Betula pendula and

physicochemical properties of its pollen at selected sites in Małopolska regiona" showed

that air pollution has a substantial impact on qualitative and quantitative results

regarding chemical compounds of pollen and on the secondary structure of Bet v1. This finding is definitely going to add new knowledge in the field of pollen allergy. However, to make manuscript more

effective I have some concerns that need to be addressed.

I. There are few typos errors throughout the manuscript that need to be revised.

II. Abstract is not structured, it is better to make it structured as per the policy of journal.

III. Figure quality and legends need to be revised for clarity.

IV. The word literature may be replaced by references as per journal guidelines.

V. References are not uniform revise it and make uniform.

overall, the manuscript is presented comprehensively and with the suggested minor changes, I recommend acceptance.

Reviewer #3: A more clear title should be inserted.

The abstract should be structured in Introduction, methods,Results, Conclusion.

The aim and novelty character of paper should be better marked.

A graphical scheme of study approach should be inserted.

Ftir raman spectroscopy results should be better discussed.

Limits, advantages and practical applications shoulbe better highlighted in Conclusion.

6. PLOS authors have the option to publish the peer review history of their article (what does this mean?). If published, this will include your full peer review and any attached files.

Reviewer #1: No

Reviewer #2: No

Reviewer #3: No

---

## [Author Response · Author response to Decision Letter 0]

30 Oct 2022

Reviewer Comments to the Author

Reviewer #1: This paper describes a study about the possible relationship between the physiological conditions of Betula pendula and the air pollution. This is a rather relevant contribution to the better understanding of the problem of increasing allergy of pollen in polluted atmospheres. A particularly important feature of this study is the acquisition of primary experimental data, which is not common. I found only one problem with the analysis of the results. Indeed, because we have a multivariate data set, and besides the univariate analysis that has been presented in the paper, a unsupervised multivariate data analysis should be done, for example principal components analysis, non-linear maps, etc. This would improve the data analysis and interpretation of the results.

Authors: Thank you very much for this suggestion. Indeed, we have a multivariate data set and we considered the possibility of performing the principal component analysis to assess the impact of the environmental pollution parameters on the physiological conditions of plants. However, the data obtained from the measuring stations contain a different number of parameters, which we noted in the manuscript text. For this reason, we don't have complete data to perform the analysis. Moreover, the parameters Fv/Fm and Fv/Fo are interdependent, so introducing them to the analysis seems to be pointless. Taking into account the slight physiological differentiation between the examined objects from different sites, we decided that such an analysis would not bring any breakthrough information to the manuscript.

Reviewer #2: The manuscript "Possible relationship between the physiological conditions of Betula pendula and physicochemical properties of its pollen at selected sites in Małopolska regions" showed

that air pollution has a substantial impact on qualitative and quantitative results regarding chemical compounds of pollen and on the secondary structure of Bet v1. This finding is definitely going to add new knowledge in the field of pollen allergy. However, to make manuscript more effective I have some concerns that need to be addressed.

Authors: Dear Reviewer, thank you very much for your kind opinion. The detailed answers are presented below. 

I. There are few typos errors throughout the manuscript that need to be revised.

Authors: The indicated errors were corrected. 

II. Abstract is not structured, it is better to make it structured as per the policy of journal.

Authors: Abstract was structured, as suggested. 

III. Figure quality and legends need to be revised for clarity.

Authors: The quality of Figures 1SM, 4 and 5 has been improved. The tiny typos were corrected.

IV. The word literature may be replaced by references as per journal guidelines.

Authors: Was done.

V. References are not uniform revise it and make uniform. 

Authors: The references list was corrected. Journal title abbreviations were used according to Medline, https://www.ncbi.nlm.nih.gov/nlmcatalog/journals

Overall, the manuscript is presented comprehensively and with the suggested minor changes, I recommend acceptance.

Reviewer #3: 

Authors: Dear Reviewer, thank you very much for your kind opinion. The detailed answers are presented below. 

A more clear title should be inserted.

Authors: The title was rearranged to be more precise.

The abstract should be structured in Introduction, methods, Results, Conclusion.

Authors: Abstract was structured, as suggested. 

The aim and novelty character of paper should be better marked.

Authors: The aim of the study was rearranged.

A graphical scheme of study approach should be inserted.

Authors: Graphical abstract was added.

Ftir raman spectroscopy results should be better discussed.

Authors: During the research, we did not perform FTIR measurements, but only FTR, so we are not able to present comparative analysis, because, if we understood the suggestion correctly, this was what the Reviewer meant. In the FTR technique, the 1064nm laser excitation was used (in the infrared range). The lowering of the energy of the laser radiation allows for the reduction of the fluorescence signal and as a consequence improves the quality of the spectrum.

These techniques (FTIR and FTR) are complementary so we haven't seen the advisability of doing both of them. The Raman spectroscopy method, using the Raman effect, successfully competes with FTIR spectroscopy, as it has a wider application and does not require drying the samples. Therefore, it would be impossible to perform FTIR measurements on liquid (aqueous) samples of Bet v1 protein due to the limitation of the IR method.

According to the Reviewer’s suggestion, we have added an additional explanation to the manuscript.

Limits, advantages and practical applications should be better highlighted in Conclusion.

Authors: A short note on practical application of the presented results was added. 

6. PLOS authors have the option to publish the peer review history of their article (what does this mean?). If published, this will include your full peer review and any attached files.

Authors: In our opinion, it is not necessary to publish the peer history of our article

---

## [Decision Letter · Decision Letter 1]

16 Nov 2022

PONE-D-22-14561R1Air pollution in the places of Betula pendula growth and development changes the physicochemical properties and the main allergen content of its pollenPLOS ONE

Dear Dr. Ziemianin,

Thank you for submitting your manuscript to PLOS ONE. After careful consideration, we feel that it has merit but does not fully meet PLOS ONE’s publication criteria as it currently stands. Therefore, we invite you to submit a revised version of the manuscript that addresses the points raised during the review process.

We look forward to receiving your revised manuscript.

Kind regards,

Umakanta Sarker

Academic Editor

PLOS ONE

Journal Requirements:

Additional Editor Comments (if provided):

Although three reviewers accepted the manuscript, it has a lot of typographical mistake that must be addressed well before getting its final acceptance.

Check the whole manuscript for typos and addressed them accurately.

Reformat the abstract following journal format.

Line 143-144 and Table 1: Write minute symbol in the place of inverted coma.

Line 213: Change "–30° C" to "–30 °C". follow this style for whole manuscript where it exists.

Line 219: Change "ml" to "mL". follow this style for whole manuscript where it exists.

Add a space between number/value and unit (mL); before and after the sysmol "="; ">"; "<".

Reviewers' comments:

Reviewer's Responses to Questions

**Comments to the Author**

1. If the authors have adequately addressed your comments raised in a previous round of review and you feel that this manuscript is now acceptable for publication, you may indicate that here to bypass the “Comments to the Author” section, enter your conflict of interest statement in the “Confidential to Editor” section, and submit your "Accept" recommendation.

Reviewer #1: All comments have been addressed

Reviewer #2: All comments have been addressed

Reviewer #3: All comments have been addressed

2. Is the manuscript technically sound, and do the data support the conclusions?

Reviewer #1: Yes

Reviewer #2: Yes

Reviewer #3: Yes

3. Has the statistical analysis been performed appropriately and rigorously? 

Reviewer #1: Yes

Reviewer #2: Yes

Reviewer #3: Yes

4. Have the authors made all data underlying the findings in their manuscript fully available?

Reviewer #1: Yes

Reviewer #2: Yes

Reviewer #3: Yes

5. Is the manuscript presented in an intelligible fashion and written in standard English?

Reviewer #1: Yes

Reviewer #2: Yes

Reviewer #3: Yes

6. Review Comments to the Author

Reviewer #1: The authors have answered satisfactorily to the referees questions. I think the paper is suitable for publication.

Reviewer #2: (No Response)

Reviewer #3: The authors have well addressed all comments and the paper is now well organized and suitable for publication.

7. PLOS authors have the option to publish the peer review history of their article (what does this mean?). If published, this will include your full peer review and any attached files.

Reviewer #1: No

Reviewer #2: No

Reviewer #3: No

---

## [Author Response · Author response to Decision Letter 1]

6 Dec 2022

Comments 

Authors: The reference list and the citations in the text were checked and corrected, if necessary. As we mentioned previously (first correction), the only one position was added (no 47), therefore, positions 47 to 93 have been moved 1 forward.

Additional Editor Comments (if provided):

Although three reviewers accepted the manuscript, it has a lot of typographical mistake that must be addressed well before getting its final acceptance.

Check the whole manuscript for typos and addressed them accurately.

Reformat the abstract following journal format.

Authors: In our opinion, the abstract is correctly prepared, according to the guidelines. The sub-titles were deleted. 

Line 143-144 and Table 1: Write minute symbol in the place of inverted coma.

Line 213: Change "–30° C" to "–30 °C". follow this style for whole manuscript where it exists.

Line 219: Change "ml" to "mL". follow this style for whole manuscript where it exists.

Add a space between number/value and unit (mL); before and after the sybmol "="; ">"; "<".

Authors: All mentioned above errors were corrected. 

Reviewers' comments:

Reviewer's Responses to Questions

Comments to the Author

1. If the authors have adequately addressed your comments raised in a previous round of review and you feel that this manuscript is now acceptable for publication, you may indicate that here to bypass the “Comments to the Author” section, enter your conflict of interest statement in the “Confidential to Editor” section, and submit your "Accept" recommendation.

Reviewer #1: All comments have been addressed

Reviewer #2: All comments have been addressed

Reviewer #3: All comments have been addressed

2. Is the manuscript technically sound, and do the data support the conclusions?

Reviewer #1: Yes

Reviewer #2: Yes

Reviewer #3: Yes

3. Has the statistical analysis been performed appropriately and rigorously?

Reviewer #1: Yes

Reviewer #2: Yes

Reviewer #3: Yes

4. Have the authors made all data underlying the findings in their manuscript fully available?

Reviewer #1: Yes

Reviewer #2: Yes

Reviewer #3: Yes

5. Is the manuscript presented in an intelligible fashion and written in standard English?

Reviewer #1: Yes

Reviewer #2: Yes

Reviewer #3: Yes

6. Review Comments to the Author

Reviewer #1: The authors have answered satisfactorily to the referees questions. I think the paper is suitable for publication.

Reviewer #2: (No Response)

Reviewer #3: The authors have well addressed all comments and the paper is now well organized and suitable for publication.

7. PLOS authors have the option to publish the peer review history of their article (what does this mean?). If published, this will include your full peer review and any attached files.

Do you want your identity to be public for this peer review? For information about this choice, including consent withdrawal, please see our Privacy Policy.

Reviewer #1: No

Reviewer #2: No

Reviewer #3: No

Authors: We are truly thankful to all reviewers for their positive comments.

---

## [Editor Report · Decision Letter 2]

16 Dec 2022

Air pollution in the places of Betula pendula growth and development changes the physicochemical properties and the main allergen content of its pollen

PONE-D-22-14561R2

Dear Dr. Ziemianin,

We’re pleased to inform you that your manuscript has been judged scientifically suitable for publication and will be formally accepted for publication once it meets all outstanding technical requirements.

Kind regards,

Umakanta Sarker

Academic Editor

PLOS ONE

Additional Editor Comments (optional):

Still, some typos are retained in the MS that must be addressed at the tie of proofreading

Follow the line number of the track changed file:

Line 151-152 and Table 1: Write the minute symbol in the place of the inverted comma.

Line 220: Change "–30° C" to "–30 °C". follow this style for the whole manuscript where it

exists.
---

## [Editor Report · Acceptance letter]

3 Jan 2023

PONE-D-22-14561R2 

Air pollution in the places of *Betula pendula* growth and development changes the physicochemical properties and the main allergen content of its pollen 

Dear Dr. Ziemianin:

I'm pleased to inform you that your manuscript has been deemed suitable for publication in PLOS ONE. Congratulations! Your manuscript is now with our production department. 

Kind regards, 

on behalf of

Professor Umakanta Sarker 

Academic Editor

PLOS ONE